# Strong and ductile high temperature soft magnets through Widmanstätten precipitates

Liuliu Han [1] ✉, Fernando Maccari [2], Ivan Soldatov [3], Nicolas J. Peter [1], Isnaldi R. Souza Filho [1], Rudolf Schäfer [3], Oliver Gutfleisch[2], Zhiming Li [4] ✉ & Dierk Raabe [1] ✉

Fast growth of sustainable energy production requires massive electrification of transport, industry and households, with electrical motors as key components. These need soft magnets with high saturation magnetization, mechanical strength, and thermal stability to operate efficiently and safely. Reconciling these properties in one material is challenging because thermally-stable microstructures for strength increase conflict with magnetic performance. Here, we present a material concept that combines thermal stability, soft magnetic response, and high mechanical strength. The strong and ductile soft ferromagnet is realized as a multicomponent alloy in which precipitates with a large aspect ratio form a Widmanstätten pattern. The material shows excellent magnetic and mechanical properties at high temperatures while the reference alloy with identical composition devoid of precipitates significantly loses its magnetization and strength at identical temperatures. The work provides a new avenue to develop soft magnets for high-temperature applications, enabling efficient use of sustainable electrical energy under harsh operating conditions.

There is an increasing demand for soft-magnetic materials (SMMs) that can be used at high operating temperatures and mechanical loads, such as in high-performance electrical machines. Such applications require alloys with reliable magnetic and mechanical performance at high thermal stability[1,2]. This is a severe material design challenge because SMMs are today tuned for multifunctional profiles to minimize energy losses at high efficiency[3], but they lose many of these features when exposed to high temperatures and stresses[4,5]. High operating temperatures and strength become essential for electrical machines for two reasons. The first one is the need to operate motors at higher rotor speeds and torques, enabling larger mechanical power output at higher efficiency. The second one refers to the exposure of electrical machines to more severe thermal and mechanical environments for using

sustainable electrification in markets that are today propelled by fossil-fueled machines[6].

According to Bloch's law, saturation magnetization decays at higher temperatures ($T$) with $T^{3/2}$ dependence due to increasing thermal fluctuations[7,8]. High-performance electrical machines experience today higher temperatures due to dissipative heat from operation and from high-temperature environments[5]. This not only reduces magnetization and, thus, efficiency but also leads to the mechanical softening of the material due to the easier surpassing of thermal barriers for dislocation motion and multiplication. The increased dissipative heat in conjunction with loss in strength under mechanically highly demanding application scenarios can thus lead to catastrophic failure of electrical motor components under such conditions[9,10]. This material design challenge is thus characterized by

[1]Max-Planck-Institut für Eisenforschung, Max-Planck-Straße 1, 40237 Düsseldorf, Germany. [2]Department of Material Science, Technical University of Darmstadt, 64287 Darmstadt, Germany. [3]IFW Dresden, Institute for Metallic Materials, Helmholtzstr. 20, 01069 Dresden, Germany. [4]School of Materials Science and Engineering, Central South University, 410083 Changsha, China. ✉e-mail: l.han@mpie.de; lizhiming@csu.edu.cn; d.raabe@mpie.de

the need for high magnetization, high mechanical strength, and low hysteresis loss, where all of these features must be retained over a wide (high) temperature range. A typical target temperature for such scenarios is as high as ~673 K at a strength above 500 MPa, yet, current high-performance magnetic materials are limited to temperatures around 600 K[5]. Thus, developing better magnetically soft, mechanically strong, and ductile materials that are stable at high temperatures is vital for realizing a sustainable and safety-critical society and industry[11].

Currently, the alloy design of such high-temperature applications is mostly based on cobalt–iron (Co–Fe), with a focus on enhancing magnetic properties at a fixed temperature (~600 K), yet, with less consideration of reconciling this with the other features mentioned[12–14]. The intrinsic limit behind this is that conventional dilute alloys cannot meet multiple functions when tapping only from a limited compositional and phase space, which reduces the degrees of freedom for realizing both, the desired magnetic functions and the mechanically required microstructures[15,16]. Co–Fe alloys based on intermetallic compounds with high Curie temperatures are capable of maintaining their magnetic moments upon thermal excitation at higher temperatures. Yet, it is hard to suppress the intrinsic brittleness of these compounds[17,18]. Therefore, the excellent thermal stability of (a) microstructure, (b) mechanical strength and damage tolerance, and (c) magnetic properties is a primary goal for the development of high-temperature SMMs.

A typical strategy for this is to use coherent nanoprecipitates to reconcile low coercivity and high mechanical strength at room temperature[3,19]. However, this approach fails at elevated temperatures due to the high capillary-driven coarsening rate of nanoprecipitates. Also, introducing coherent nanoprecipitates requires chemical composition adjustment, which involves the use of non-ferromagnetic elements, thus reducing magnetization and motor efficiency.

For a new alloy design approach, we therefore identified four main rules to meet the described multi-functionality demands when exposing SMMs to harsh thermal and mechanical environments. First, the chemical composition and associated intrinsic magnetic properties determine the resistance against temperature-induced magnetization loss. Hence, a high concentration of strong ferromagnetic elements with the highest possible Curie temperature should be used. Second, intrinsically strong and thermally stable precipitates with well-tunable size and number density should be inserted to enable and maintain the mechanical strength of the alloy over a wide temperature range without embrittling it. Third, diffusion-controlled growth and chemical partitioning kinetics among phases must be suppressed to maintain magnetic moment and strength. Fourth, all microstructural features must exert minimal pinning forces on magnetic domain walls to keep coercivity low. This means that the micromagnetically relevant length scales of the different phases should be larger or smaller than the width of the magnetic domain walls.

We have translated these constraints into a general alloy design concept by triggering intermetallic precipitates arranged in a Widmanstätten-pattern[20] with a large aspect ratio in a ferromagnetically soft matrix with high magnetization. Along the design criteria outlined above the Widmanstätten-patterned precipitates meet several requirements: (1) both the matrix and the precipitates are ferromagnetic; (2) their precipitation from the ferromagnetic matrix does not harm the magnetization of the matrix (e.g., by depleting it from ferromagnetic elements via partitioning); (3) precipitate dimensions, shape factors, and patterning are below or above the limits of magnetic domain wall pinning, respectively; (4) precipitates act as additional nucleation sites for domain wall nucleation and growth (for good soft magnetic response); (5) the Widmanstätten-patterned arrangement forms a strong barrier against dislocation motion (equipping the material with mechanical strength); (6) precipitates maintain magnetic and mechanic stability at high temperatures.

These considerations have been realized in a non-equiatomic iron–nickel–cobalt–tantalum ($Fe_{35}Co_{30}Ni_{30}Ta_5$ (at%))[19] multi-component material, exploiting many degrees of freedom in compositional adjustment enabled by the high-entropy alloy concept[21–24]. The new alloy design rationale offers access to a property spectrum that has so far been untapped, where soft magnetic features can be seamlessly combined with strength, ductility, and thermal stability.

## Results and discussion

### Alloy design and microstructure analysis

We conducted thermodynamic (chemical composition, crystal structure) and kinetic (diffusion coefficients) simulations to realize the new material design concept with two coexisting phases (matrix, Widmanstätten precipitates) in the $Fe_{35}Co_{30}Ni_{30}Ta_5$ (at%) alloy system using the software Thermo-Calc (Fig. 1a). The target for these two coexisting phases was a strong ferromagnetic solid solution matrix with high crystal symmetry (enabling ductile plastic response) forming the major fraction and an intermetallic phase with a lower crystal symmetry (enabling resistance to plastic flow) with the potential to form a Widmanstätten pattern (to meet the morphological constraints described above). The simulation results (Fig. 1a) show that a $D0_a$ phase nucleates from the face-centered cubic (fcc#1) phase below 1360 K for this alloy composition. The $D0_a$ crystal structure is an ordered $D0_{19}$ structure based on a hexagonal crystal lattice. The fcc#1 phase further decomposes into a body-centered cubic phase and another fcc#2 phase below 1015 K. Therefore, we select a higher temperature, i.e., 1173 K, for isothermal heat treatment to obtain a dual-phase structure. We synthesized the homogenized multicomponent alloy (denoted as HM-MCA) with a single fcc crystal structure by conventional hot-rolling and homogenization of the cast alloy ingot (see "Methods"). The Widmanstätten-patterned multicomponent alloy (referred to as WP-MCA) with well-controlled precipitate size and number density was achieved after further isothermal heat treatment of the HM-MCA at 1173 K for 2 h. Figure 1b shows the X-ray diffraction (XRD) patterns of the WP-MCA, with two families of diffraction peaks (face-centered cubic and hexagonal crystal). The microstructure has been probed by electron channeling contrast imaging (ECCI) (Fig. 1c). After annealing, a high number density of precipitates ($6.2 \times 10^{16}$ m$^{-3}$) with typical Widmanstätten pattern[20]) is observed within a coarse-grained matrix in the WP-MCA (average grain size of $90.7 \pm 38.2$ μm, vs. HM-MCA ~$64.8 \pm 9.8$ μm[19], Fig. S1a). The phase fraction obtained by Rietveld refinement[25] of the XRD results (cubic ~93%, hexagonal ~7%) matches well with the experimentally observed ones by averaging over multiple ECC images (fig. S1b). The transmission Kikuchi diffraction (TKD) phase map (Fig. 1d) confirms the precipitates and matrix as hexagonal and fcc crystal structures, respectively. Fig. S1c shows the bright-field transmission electron microscopy (BF-TEM) image pertaining to the region denoted by a yellow frame in the TKD map (Fig. 1d). The corresponding dark-field (DF) image (Fig. 1e) is acquired using the precipitate reflection of the selected area electron diffraction (SAED) pattern (Fig. S1d). The atomic-resolution scanning TEM (STEM) micrograph of the phase boundary shows that the precipitates are incoherent with the fcc matrix (Fig. 1f). The precipitates assume an ordered $D0_{19}$ structure as demonstrated by the corresponding fast Fourier transformation (FFT) image (Fig. 1g) and the SAED data (Fig. S1d). Figure 1h–j shows the three-dimensional (3D) morphology and elemental distributions of the constituents in the WP-MCA material down to the near-atomic scale, mapped by scanning electron microscopy (SEM) and atom probe tomography (APT). The SEM image of an APT sample prior to sharpening reveals the plate-shaped precipitates (Fig. 1h). The APT reconstruction in Fig. 1i and the corresponding cross-sectional tomogram of a 2.5 nm thick plate show distinct chemical differences between the fcc matrix and the $D0_{19}$ precipitate. Figure 1j shows a one-dimensional (1D) compositional profile obtained along the arrow in Fig. 1i. These results reveal Ta and Ni enrichments in the

$DO_{19}$ precipitates and Fe enrichment in the fcc matrix, while the Co content is almost equi-partitioned in the two phases. The average chemical compositions of the fcc and $DO_{19}$ phases, obtained from three APT data sets, were $Fe_{39}Co_{31}Ni_{28}Ta_2$ and $Ni_{46}Co_{29}Ta_{20}Fe_5$ (at%), respectively.

## Magnetic properties at room and elevated temperatures

To show the excellent resistance against loss in magnetization and coercivity increase when exposed to a wide (high) temperature range (300–873 K) of the WP-MCA material, we studied the same features also for the alloy variant with identical chemical composition but precipitate-free microstructure (HM-MCA). Figure 2 shows the bulk magnetic data over a wide temperature range of 5–873 K measured by vibrating-sample magnetometry (VSM) under open-circuit conditions. Both materials exhibit typical soft-ferromagnetic behavior up to 873 K according to the hysteresis loops (Fig. S2). The current WP-MCA

material shows isotropic magnetic performance (Fig. S3a) due to the polycrystalline grain structure. Figure 2a, b shows the saturation magnetization ($M_s$) and coercivity ($H_c$) values averaged from three hysteresis loop measurements under open-circuit conditions. We corrected the demagnetization effect based on the shape of the specimens. The demagnetization slightly changes the shape of the loop but does not notably affect the values of $M_s$ and $H_c$ (Fig. S3b–d). The magnetic performance of the WP-MCA at room temperature (300 K) was also measured under closed-circuit conditions (Fig. S3e). The results are comparable with the values achieved in an open-circuit condition experiment, indicating that the values achieved from the VSM measurement are precise. The $M_s$ values of the WP-MCA material at 5 K and 300 K are comparable to those of the HM-MCA material, showing the intrinsic origin of $M_s$. More importantly, when exposed to higher temperatures, the precipitation-containing alloy WP-MCA shows excellent maintenance of its magnetization. More specifically,

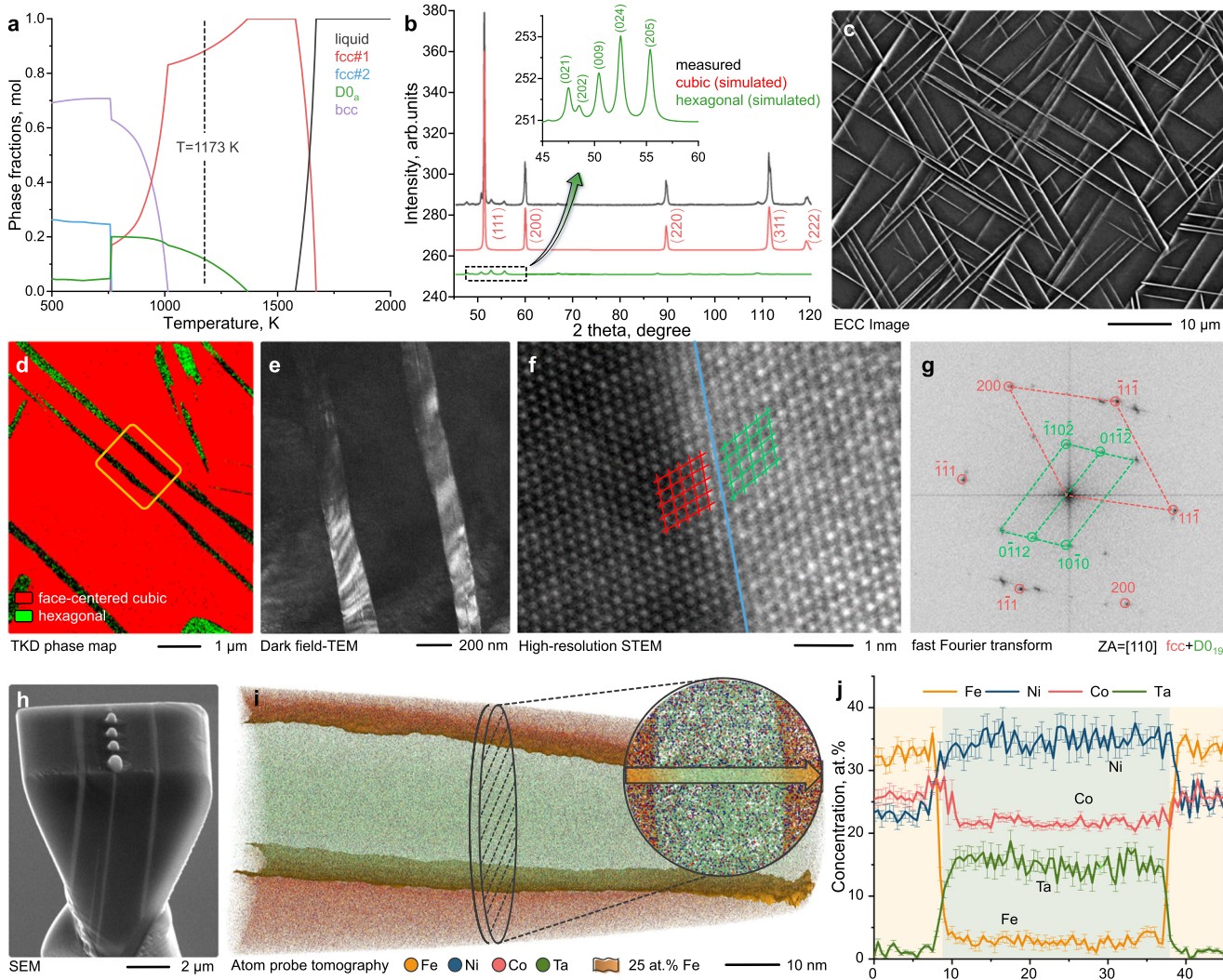

**Fig. 1 | Microstructure analysis. a** Calculated equilibrium phase diagram of the $Fe_{35}Co_{30}Ni_{30}Ta_5$ (at%) alloy using the thermodynamics software Thermo-Calc (see "Methods"). The $DO_a$ structure is an ordered $DO_{19}$ structure based on a hexagonal crystal lattice. The black dashed line indicates the chosen isothermal annealing temperature (1173 K). **b** XRD patterns (measured and simulated). The inset shows the enlarged segment between 45° and 60° (black dashed frame). **c** ECCI analysis showing the microscale Widmanstätten-patterned precipitates. **d** TKD phase map showing the identified phases. The yellow frame refers to the region shown in (**e**) and Fig. S1c. **e** DF-TEM image taken using $DO_{19}$ reflection (Fig. S1d). **f** Atomic-resolution STEM micrograph showing the incoherent phase boundary. Left side, fcc matrix; Right side, $DO_{19}$ precipitates. **g** Corresponding FFT pattern. Red, fcc matrix, green, $DO_{19}$ precipitates. **h** An APT sample before sharpening showing the 3D plate-shaped precipitates. **i** Reconstructed the APT map showing the elemental distributions. The interface is highlighted with 25 at% Fe. The plane view (right-top corner) shows a 2.5 nm-thick cross-sectional slice from the tip. **j** 1D compositional profiles along the arrow marked in the plane view. The error bars are estimated as described in Methods.

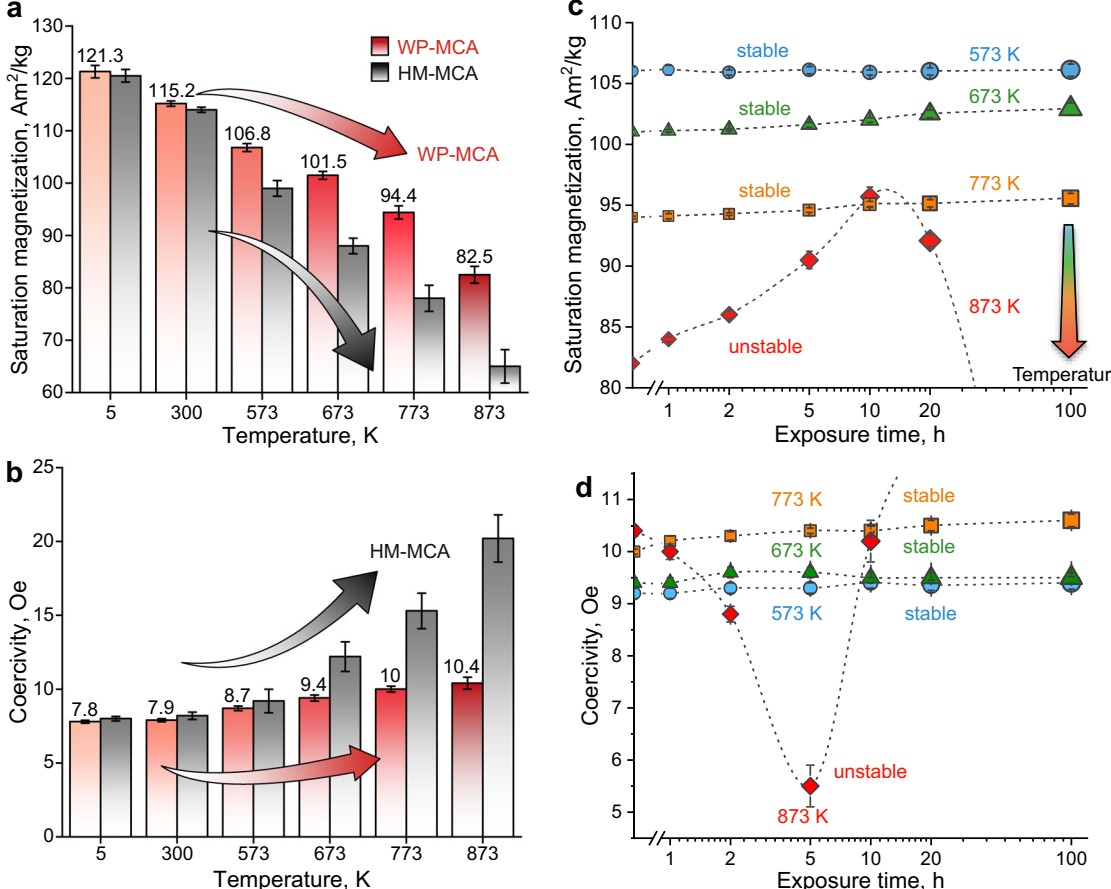

**Fig. 2 | Magnetic properties and thermal stability at elevated temperatures.** **a** Saturation magnetization and **b** intrinsic coercivity acquired from the bulk hysteresis loop measurement over a wide temperature range. **c** The evolution of the saturation magnetization and the **d** Intrinsic coercivity as a function of exposure time at elevated temperatures for the new alloy WP-MCA. All standard deviations are obtained by at least three measurements.

the $M_s$ values are 107, 102, 94, and 83 Am$^2$/kg at 573, 673, 773, and 873 K, respectively. In contrast, a pronounced loss in $M_s$ is observed for the single-phase HM-MCA reference alloy with increasing temperature, a feature also commonly observed in conventional SMMs[3]. The WP-MCA material maintains 94, 89, 83 and 73% of its room-temperature $M_s$, i.e., 7%, 15%, 21%, and 25% above the values found for alloy HM-MCA. The maintenance in magnetization of alloy WP-MCA is comparable to that of the conventional Co–Fe (90.7 wt% Co) alloy, which, however, contains a forbiddingly high ratio of cobalt required for its high Curie temperature[26]. The saturation induction ($B_s$) of WP-MCA is 1.1 T at 773 K, while the $B_s$ value of the 90.7 wt% Co alloy is between 0.8 T (923 K) and 1.2 T (523 K)[26]. We also compared the $B_s$ value of the current MCAs with a wide range of commercial SMMs in an overview plot (Fig. S4a). It is important to note that the new alloy's good maintenance of its magnetization at high temperatures is achieved at no expense of coercivity (Fig. 2b). Compared to the single-phase HM-MCA with a notable increase in $H_c$ towards higher temperatures from 8.2 Oe at 300 K to 20.2 Oe at 873 K, the $H_c$ values of the WP-MCA material only slightly increase with temperature but stay in the range of the values found for other SMMs (<13 Oe): the $H_c$ values are 8.7 Oe (573 K), 9.4 Oe (673 K), 10.0 Oe (773 K), and 10.4 Oe (873 K), respectively, for the new WP-MCA material.

To gain further insights into the mechanisms and influence of the microstructural and compositional changes on the thermal magnetic stability of the materials under elevated operating temperatures, we isothermally processed the WP-MCA material at 573–873 K for different exposure times (0–100 h) and probed the magnetic properties at these temperatures. The average $M_s$ (Fig. 2c) and $H_c$ (Fig. 2d) values

remain almost unchanged after annealing at 573, 673, and 773 K for up to 100 h. Further increasing the annealing temperature (873 K) leads to interesting phenomena: (1) $M_s$ increased with prolonged annealing up to 10 h, that is, from 82.1 Am$^2$/kg (0 h) to 90.5 Am$^2$/kg (5 h) and to 95 Am$^2$/kg (10 h); (2) $H_c$ first decreases from 10.4 Oe (0 h) to 5.5 Oe (5 h) and then increases again to 10.2 Oe (10 h). This can be attributed to the formation of the nanoscale Fe-enriched segregations observed to occur during annealing, as revealed by the elemental distribution and statistical deviation analysis (Fig. S5). The results show that alloy WP-MCA has excellent thermal stability up to 773 K, indicating its excellent potential for high-temperature applications.

Next, we unravel the mechanisms responsible for the excellent resistance to magnetization loss at high temperatures. The WP-MCA material shows a higher Curie temperature (933 K) than the HM-MCA (907 K) according to the thermomagnetic and differential scanning calorimetry (DSC) heating curves (Fig. 3a), owing to its stronger ferromagnetic matrix and the resulting higher exchange interaction. We cast and measured a model alloy (to serve as a reference material regarding the effect of the precipitates) with a nominal composition identical to that of the fcc matrix (Fe$_{39}$Co$_{31}$Ni$_{28}$Ta$_2$, at%) of the WP-MCA using the composition acquired by APT analysis. Figure 3b shows the $M_s$ values of the nominal fcc phase at room and elevated temperatures measured from the hysteresis loop (Fig. S6a) and the corresponding DO$_{19}$ precipitate ones by calculation (Methods). The $M_s$ for the nominal fcc phase decreased by 29% with increasing temperature from 300 to 873 K, while the decrease for the DO$_{19}$ precipitate is only 13%. Figure S6b summarizes the $M_s$ values of each phase as a portion of the total magnetization in the WP-MCA at different temperatures. The $M_s$ of the

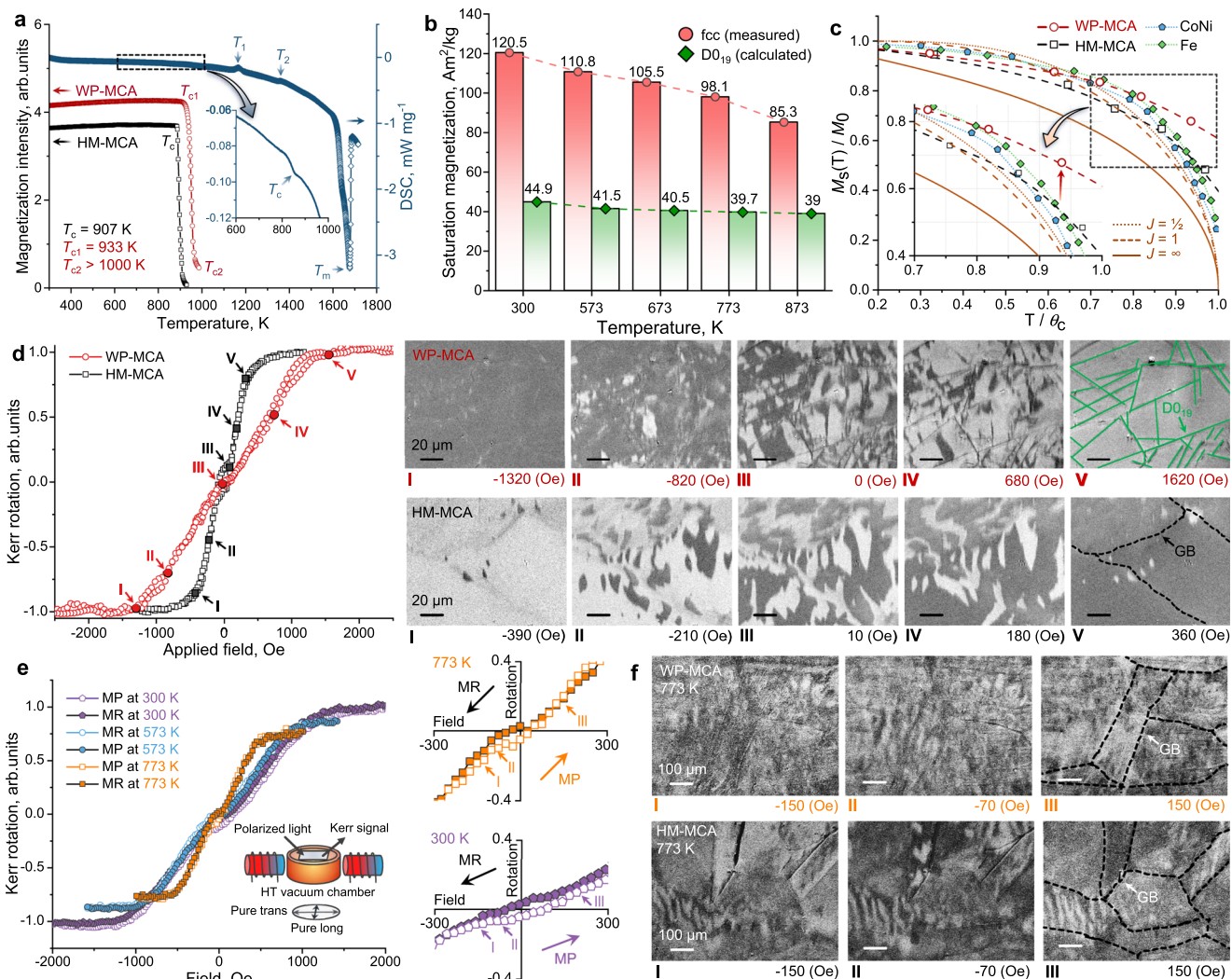

**Fig. 3 | Magnetic performance and domain wall pinning mechanisms at room and elevated temperatures. a** Thermomagnetic and DSC curves. $T_{c1}$ and $T_{c2}$ on the thermomagnetic curve of the WP-MCA denote the Curie temperatures ($T_c$) of the fcc matrix and $D0_{19}$ precipitates, respectively. $T_1$ and $T_2$ on the DSC curve show the exothermic and second-order effects, respectively. **b** $M_s$ value of the fcc matrix and the $D0_{19}$ precipitate at different temperatures. **c** Temperature dependence of reduced saturation magnetization. The Brillouin function theory expressions with $J = 1/2, 1$ and $\infty$ are shown, where $J$ represents the total angular momentum quantum number of the atom. The experimental results are compared for the current new

alloys, cobalt-nickel, and pure iron as reference materials. **d** Evolution of domain structures following the hysteresis loop (left) under Kerr observation at room temperature. The schematic illustrates the $D0_{19}$ precipitates in the WP-MCA and the grain boundary (GB) in the HM-MCA. **e** Magnetization as a function of the applied external magnetic field acquired from the surface loop by high-temperature MOKE measurement of the WP-MCA. MP magnetization process, MR magnetization reversal. The corresponding magnified loop curves at 773 K (right-top) and 300 K (right-bottom) are shown. **f** The magnetic domain structure evolution of the WP-MCA (top images) and HM-MCA (bottom images) at 773 K, respectively.

$D0_{19}$ precipitates shows an increased contribution (40%) to the magnetization from 300 K to 873 K, indicating good resistance to the temperature-induced reduction of the magnetization. The evolution of the intrinsic magnetic properties (saturation magnetization and Curie temperature) is quantitatively analyzed using the Brillouin theory (Methods). Figure 3c shows the $M_s(T)/M_s(0)$ values regarding the $T/T_c$ of the current MCAs and several commercial soft magnets. This comparison shows that the $M_s(T)/M_s(0)$ values of the WP-MCA are higher than those of the HM-MCA, pure iron, and Co-Fe towards higher temperatures ($T/T_c > 0.7$)[27].

The current strategy of introducing microscale precipitates, which is an unusual feature for soft magnets, avoids a significant increase in coercivity while maintaining good high-temperature magnetization. To understand the effect of the precipitates on magnetization and magnetization reversal process, we investigated the evolution of the magnetic domain structure of the current MCAs by wide-field magneto-optic Kerr effect (MOKE) microscopy (Fig. 3d).

This is motivated by the fact that the nucleation and growth of magnetic domains in bulk SMMs are crucial in determining coercivity. More specifically, the nucleation of magnetic domains in the precipitate-free HM-MCA at room temperature mainly occurs at the grain boundaries (Fig. 3d), which act as heterogeneous nucleation sites. The subsequent domain growth proceeds via domain wall motion until the overall magnetization is aligned in the same direction (Fig. 3d, right-bottom, II–IV). The observed surface domain structure is typical for bulk SMMs with cubic anisotropy (for example, Fe–Ni and Fe–Si alloys[28]). The slight variations are derived from the individual crystallographic orientations of the constituting grains. In addition to grain boundaries, the phase boundaries act as additional nucleation sites in the WP-MCA (Fig. 3d, right-top, I–II). The subsequent domain growth proceeds via domain wall motion (Fig. 3d, right-top, III–V). When considering that the magnetization loops measured on the surface are less relevant for bulk materials[29], it should be noted that the observed larger coercivity field can be attributed to the redistribution

of the magnetic domains during flux change in the volume below the surface.

We also observed a temperature-dependent magnetization behavior, as indicated by the gradual increase in magnetic susceptibility with increasing temperature from 300 to 773 K (Fig. 3e). The evolution of magnetic domains in the WP-MCA at high temperatures (Fig. 3f) remains fundamentally the same as at room temperature, indicating its high thermal stability. The domain switching of the current MCAs at both room and high temperatures is shown in the Supplementary Videos. Because coercivity is a microstructure-sensitive property determined by the specific interactions between magnetic domain wall motion and various types of lattice defects, three main mechanisms are supposed to explain the excellent maintenance of the coercivity of alloy WP-MCA over such a wide temperature range (300–773 K). First, grain coarsening reduces the effect of grain boundary pinning on the domain wall movement and thus decreases coercivity according to the theory on grain size dependence of coercivity (GSDC)[30]. This is based on the fact that the measured critical single-domain size (WP-MCA, $2.1 \pm 1.9\,\mu m$) is far below the average grain size ($90.7 \pm 38.2\,\mu m$). Therefore, the theoretical reduction in coercivity of the WP-MCA material when only considering the grain size difference compared to that of the HM-MCA is only 0.9 Oe at 300 K. This value is estimated by interpolating the grain size from the alloy variants with identical composition in single phase structure but different homogenization times, thus different grain sizes (Fig. S7). Second, partial removal of dislocations during annealing leads to a lower strain field barrier for the domain wall movement, further reducing coercivity. This is because the interaction volume of dislocations is comparable to the domain wall width, leading to a pinning effect on the domain wall. The dislocation density of alloy WP-MCA is $-(2.9 \pm 0.7) \times 10^{12}\,m^{-2}$, which is lower than that of the HM-MCA material with $-(5.6 \pm 1.3) \times 10^{12}\,m^{-2}$ at 300 K (Fig. S8). This effect is expected to be more prominent due to the removal of dislocations via cross slip, climb, and annihilation at higher temperatures. Third, the reduced local lattice distortion contributes to lower coercivity. More specifically, the fcc phase in the WP-MCA material has a weaker pinning effect on the domain wall motion than that of the alloy HM-MCA. This is because the relatively large Ta atoms (146 pm) are partially replaced by Fe (126 pm) and Co (125 pm) in the fcc matrix due to the elemental partitioning during annealing. Therefore, the internal stress level and the associated strain field decrease, reducing magnetic pinning. This is proposed to explain the good coercivity of the WP-MCA material at elevated temperatures according to the high-temperature XRD measurement (Fig. S9). Weaker lattice distortion would increase the intensity of the diffraction peaks due to the scattering effect of lattice distortions[31,32]. More specifically, the full width at half maximum (FWHM) value of the (200) peak in the alloy WP-MCA at 773 K decreased by 4.3% compared to its room temperature value and is two times larger than that for the HM-MCA material. The elemental partitioning due to the precipitation also results in an enhanced Curie temperature of alloy WP-MCA (Fig. 3a), suggesting an increase in exchange interaction. Although the microscale precipitates with incoherent boundaries lead to domain wall pinning, which is detrimental to the coercivity, this effect is counterbalanced by partially releasing the structural defects, e.g., grain boundaries, dislocations, and lattice distortion. Owing to the elemental partitioning, the stronger magnetic matrix phase with enhanced intrinsic properties is an additional factor for the excellent magnetic performance of the material.

## Mechanical properties at room and elevated temperature

Our design concept of triggering incoherent intermetallic precipitates in the form of Widmanstätten patterns enables a considerable strengthening effect over a wide temperature range (300–873 K). This is important for applications where the material gets exposed to severe mechanical loads under harsh thermal conditions. Figure 4a shows the average mechanical yield strength ($\sigma_y$) of alloys WP-MCA and HM-MCA

at room and elevated temperature averaged from three tensile tests. The $\sigma_y$ value of the WP-MCA (640 MPa) is 28% higher than that of the HM-MCA material (501 MPa) at room temperature. The alloy WP-MCA contains microscale intermetallic precipitates and has good ductility of 37% under tensile loading conditions (Fig. 4b). This is due to its high work-hardening rate, which counteracts brittle failure (Fig. S10). The WP-MCA material shows remarkable resilience against high-temperature softening up to 873 K. In contrast, a progressive decrease in $\sigma_y$ is observed with increasing temperature for the HM-MCA reference material (Fig. 4a). The $\sigma_y$ values and the percentage of maintained room-temperature strength for the WP-MCA at 773 K are 587 MPa and 92%. In contrast, the corresponding values for HM-MCA are only 238 MPa and 48%, respectively. A diagram comparing the temperature dependence of the yield strength of the current WP-MCA with several commercially available high-temperature SMMs is shown in Fig. S4b. Notably, the $\sigma_y$ value of the WP-MCA is 3 times larger than that of the conventional FeCo-2V[33] alloy which has a $\sigma_y$ value of 204 MPa at 300 K and 183 MPa at 873 K, respectively. The inset in Fig. 4b shows an overview of the fracture surface of the WP-MCA material after tensile testing at room temperature. Two types of fracture morphology are identified in the enlarged views (Fig. 4c), namely, quasi-cleavage morphology with cleavage steps along the interfaces between the Widmanstätten precipitates and the adjacent fcc matrix (top image) and ductile fracture with well-defined dimples in sub-micron size in some fcc grains (bottom image). This observation indicates that the surrounding ductile matrix compensates for the intrinsic brittleness of the intermetallic precipitates. To further understand the deformation behavior, we studied the cross-sectional deformation microstructure using geometrically necessary dislocation (GND) analysis. This has been done using the local lattice rotations quantified via Kernel average misorientation maps obtained from the EBSD and ECCI data at different hardening stages[34]. Dislocation pile-ups at phase boundaries are observed at the early deformation stage (local strain, $\varepsilon = 10\%$) (Fig. 4d$_I$, e$_I$). Further straining ($\varepsilon = 30\%$) leads to dislocation accumulation along the $\{111\}_{fcc}$ plane trace (Fig. 4e$_{II}$). The dislocation arrays and pile-up features lead to a high dislocation density value ($-5 \times 10^{17}\,m^{-2}$), particularly at the phase boundaries (Fig. 4d$_{II}$). With a further increase of $\varepsilon$ to 50% (Fig. 4d$_{III}$), crystallographically aligned microbands in the fcc matrix start to shear the precipitates. Accordingly, microcracks are generated at the precipitate intersections (Fig. 4e$_{III}$). The gradually increased pile-up configurations cause high local stress peaks and enhance strain hardening, as confirmed by the quantification of the local misorientation analysis (Fig. S11). The same measurements were conducted as a reference for the material in its strain-free state. A negligible misorientation value (0.2°) is identified adjacent to the phase boundary compared to those of the alloy in its strained states (for example, 8.7° at $\varepsilon = 50\%$). Therefore, the increase in GNDs during straining is attributed to the high deformation mismatch across the matrix-precipitate hetero-interfaces.

Further, we investigated the deformation substructures after tensile tests at higher temperatures, i.e., 673 K and 773 K. The WP-MCA material shows a similar deformation behavior as at room temperature (Fig. S12c–f), in which the Widmanstätten-patterned precipitates provide a strong barrier against the movement of dislocations (Fig. 4d, e). Regular fcc dislocation slip without build-up of GNDs is the prevalent deformation mechanism for the HM-MCA material (Fig. S12i–l), similar to what is observed for many other fcc alloys[35]. However, a transition of the internal damaging mechanism from transcrystalline to intercrystalline occurred in the HM-MCA material with increasing temperature from 673 K to 773 K (Fig. S12g, h). In contrast, no grain boundary embrittlement was found in the WP-MCA at all evaluated temperatures (Fig. S12a, b), indicating its remarkable thermal stability. These observations show that the D0$_{19}$ intermetallic Widmanstätten-pattern precipitates are essential for maintaining the mechanical properties of the WP-MCA, both at room temperature and particularly

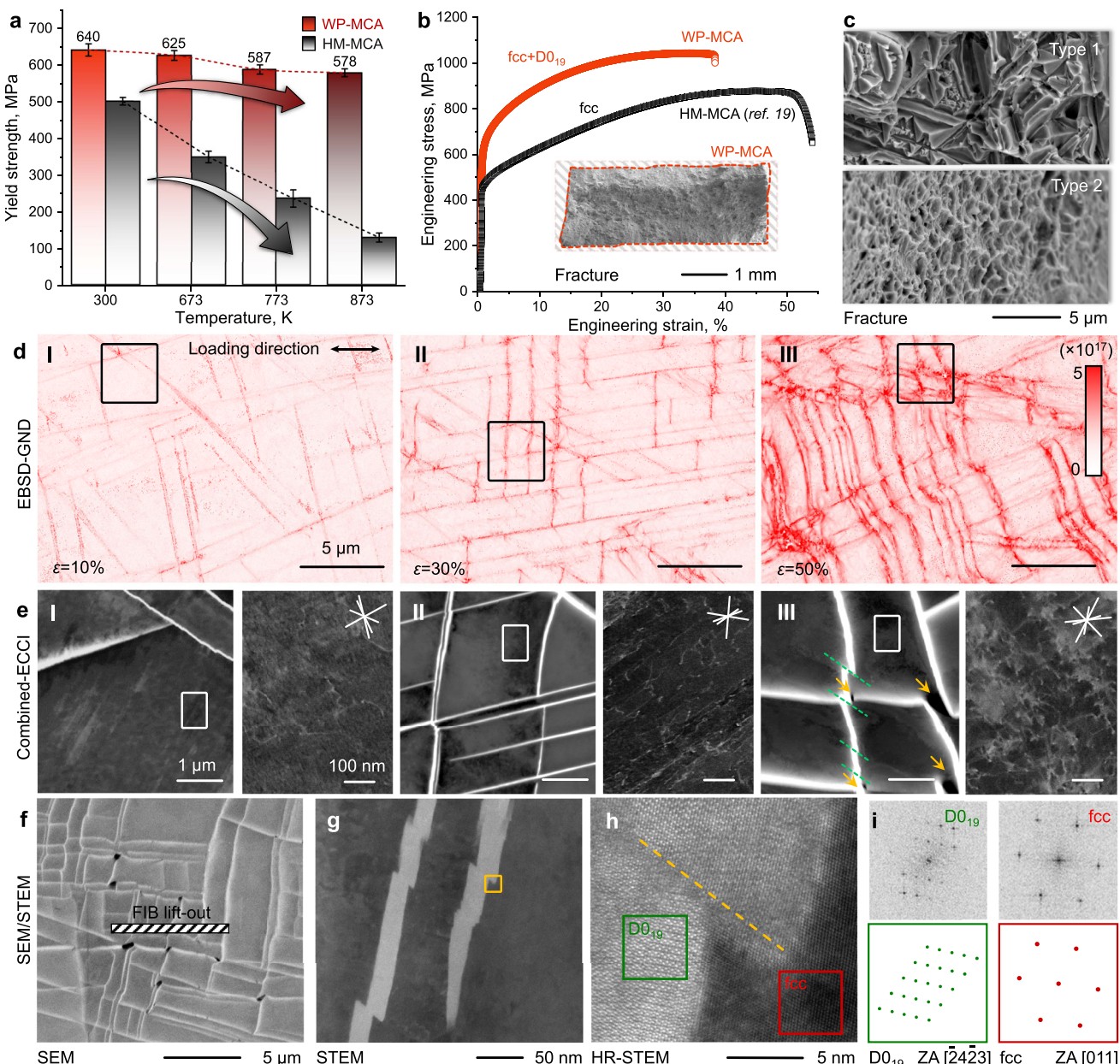

**Fig. 4 | Mechanical behavior of the WP-MCA. a** Room and high-temperature yield strength quantified by tensile experiments. For comparison, the values of the single-phase HM-MCA are also shown. The standard deviations were obtained by at least three measurements. **b** Typical engineering stress-strain curves measured at room temperature. The inset shows the overall fracture after the tensile test (dash frame region) and the area reduction. **c** Enlarged view of two typical fracture morphologies. **d** Microstructural variations as a function of deformation strain showing the accumulation of deformation-induced GNDs at the phase interface. **e** Correlative ECC images taken from the black frames in the GNDs maps showing the sheared precipitates and observable microcracks formed during deformation.

The shearing of the precipitates is highlighted by green dashed lines, and orange arrows highlight the microcracks; High-resolution ECC images taken from the region marked by the white frames show an increase in dislocation density at higher deformation. **f** Cross-sectional SEM image showing the severely deformed and sheared $D0_{19}$ precipitates near the fracture region. The inset shows a TEM lamellar lift-out by FIB for the following observations. **g** Low magnification STEM view showing the sheared $D0_{19}$ precipitates embedding in the fcc matrix. **h** Atomically resolved HAADF−STEM micrographs (orange frame in (**g**)) of the fcc/$D0_{19}$ interface. The orange dashed line indicates a sheared interface. **i** Corresponding FFT patterns of the $D0_{19}$ precipitates (left images) and fcc matrix (right images).

also at high temperatures, due to the excellent thermal stability of the precipitates. The high thermal stability of the two coexisting phases in the WP-MCA material is also confirmed by exposing the materials at 1173–1473 K for a duration of up to 6 h (Fig. S13). No obvious precipitation coarsening and hardness change was observed. The formation of microcracks during straining is caused by (i) high localized strain induced by the non-uniform deformation between the matrix and precipitates (Fig. $4d_{II}$, $e_{II}$), owing to the insufficient number of independent slip systems inside of the precipitates;[36] and (ii) by the intrinsic hard intermetallic precipitates which lead to high pile-up

stresses at the hetero-interfaces, particularly at their intersection points (Fig. $4d_{III}$, $e_{III}$). The schematic diagrams summarize these microprocesses at different plastic and hardening stages (Fig. S14). We studied the micro- and atomic structure near the fracture surface of the WP-MCA after tensile testing at room temperature via TEM lamella lift-out. In these zones, the $D0_{19}$ precipitates have a higher tendency to be sheared (Fig. 4f). Figure 4g shows the high-angle annular dark-field (HAADF)-STEM image of the sheared $D0_{19}$ precipitates. The enlarged view (orange frame in Fig. 4g) indicates the atomic mismatch between the sheared fcc/$D0_{19}$ interface (Fig. 4h). This is further confirmed by

the FFT patterns (Fig. 4i) and the coincidence site lattice analysis (Fig. S15).

In summary, we developed a strong and ductile soft magnetic material with excellent thermal stability from room temperature up to high temperatures (e.g., 773 K) by introducing ferromagnetic Widmanstätten-patterned intermetallic precipitates into a ferromagnetic multicomponent alloy matrix. In contrast to the design approach adopted for conventional soft magnetic materials so far, where microstructural features are usually avoided (for reducing domain wall pinning), we have introduced thermally stable incoherent precipitates here. The precipitates have a large aspect ratio and exert marginal pinning on magnetic domain wall motion. They help to maintain the high magnetization of the material due to the negligible inter-phase partitioning over a wide temperature range and act as effective and thermally stable barriers against dislocation slip up to 773 K. The new design strategy enabled us to develop a material with an outstanding multi-functional property profile and maintain it when the alloy undergoes long-term exposure to harsh thermal and mechanical conditions, a feature where conventional soft magnetic materials fail to function. Future efforts on designing strong and ductile soft magnetic MCAs for high-temperature applications could target higher saturation induction and lower coercivity while maintaining their outstanding mechanical performance.

## Materials

A non-equiatomic FeCoNiTa MCA ingot of 1 kg with nominal composition $Fe_{35}Co_{30}Ni_{30}Ta_5$ (at%) was synthesized by melting and casting in a vacuum induction furnace using pure metals as ingredients (purity above 99.8 wt%) under argon atmosphere. The as-cast ingot was hot-rolled at 1473 K to an engineering thickness reduction ratio of 50% (from 10 to 5 mm). Subsequently, the alloy sheet was homogenized at 1473 K for 10 min and quenched in water. The homogenized alloy is denoted as HM-MCA ("HM" stands for "homogenized"). Detailed information on the microstructure, soft-magnetic, and mechanical properties of the HM-MCA can be found in ref. 19. The bulk chemical composition of the alloy was determined by wet-chemical analysis as $Fe_{34.5}Co_{30.3}Ni_{30.2}Ta_{5.0}$ (at.%). To obtain evenly distributed precipitates in a Widmanstätten arrangement, the HM-MCA was exposed to isothermal annealing at 1173 K for 2 h, followed by water quenching. The resulting material contains Widmanstätten-patterned precipitates and hence is referred to as WP-MCA ("WP" stands for "Widmanstätten pattern"). To investigate the thermal stability of the WP-MCA, further annealing was conducted at 573 K, 673 K, and 773 K from 1 h up to 100 h, followed by water quenching. To examine the effect of grain size on coercivity, some samples of the HM-MCA were further heat-treated at 1473 K for 2 h, 4 h, and 6 h, respectively, to achieve different grain sizes, followed by water quenching. In addition, a bulk ingot of 50 g, with identical composition to the fcc matrix phase ($Fe_{39}Co_{31}Ni_{28}Ta_2$, at %) in the WP-MCA, was obtained by arc melting. The composition of the fcc matrix phase in the WP-MCA was determined by APT. The casting was processed under the protection of the Ar atmosphere and the ingot was remelted five times to ensure chemical homogeneity.

## Methods

### Microstructural characterization

The structure of the WP-MCA was characterized using multiple probing techniques. Room-temperature X-ray diffraction (XRD) measurements were performed with an X-ray diffraction instrument (D8 Advance 25-X1) equipped with a Co $K_{\alpha 1}$ radiation source (wavelength = 1.78897 Å), operated at 40 kV and 30 mA. High-temperature XRD measurements were conducted with the diffractometer (Rikaku) equipped with a Cu $K_{\alpha 1}$ radiation source (wavelength = 1.54059 Å), operated at 45 kV and 200 mA. The heating and cooling rate is 30 K/min. 5 min helium gas with 1.5 bar and a stable helium atmosphere with 13.5 bar were applied

to avoid oxidation at high temperatures. The simulated XRD patterns and fundamental parameter fit were analyzed and refined using the Rietveld method. ECCI analysis was conducted in a Zeiss–Merlin SEM at 30 kV. TKD measurement was performed in the same SEM using the Bruker software for pattern analysis (Esprit 2.1). Electron backscatter diffraction (EBSD) measurements were conducted in a Zeiss-Crossbeam (XB 1540) focus ion beam (FIB) SEM operated at 15 kV. Phase transition temperatures were determined by differential scanning calorimetry (DSC) (NETZSCH DSC 404 C). DSC measurements were performed from room temperature to 1450 °C at a heating rate of 10 °C/min. Transmission electron microscopy (TEM) analysis was conducted under both bright-field (BF) and dark-field (DF) imaging modes in a JEOL JEM 2100+ microscope operated at 200 kV. Scanning transmission electron microscopy (STEM) micrographs were taken using a probe-corrected transmission electron microscope (Titan Themis 60-300) at 300 kV. HAADF images (convergence angle ~24 mrad) were acquired to modify the Z-contrast characteristics of the imaging mode. The collection angle ranges from 75 mrad to 200 mrad. APT experiments were performed using a local electrode atom probe instrument (Cameca Leap 5000 XR) in laser-pulsing mode with a pulse repetition rate of 125 kHz, a test temperature of 60 K, and a pulse fraction of 20%. The APT data sets were reconstructed and analyzed using the Cameca software (AP suit 6.0). The error bars in the APT analysis (1D compositional profile) were calculated by $2\sigma = \sqrt{\frac{C_i(1-C_i)}{N}}$, where $C_i$ is the composition of each solute $i$, $N$ is the total number of atoms of the analysis volume. The TEM lamellas and APT tips were prepared with the dual FIB-SEM system (FEI Helios Nano-Lab 600i).

### Mechanical testing

Flat dog-bone-shaped specimens with a gauge length of 30 mm, a total length of 60 mm, a gauge width of 5 mm, and a thickness of 2 mm were cut by electron discharging machining along the rolling direction of the WP-MCA. Uniaxial tensile tests were conducted with an initial strain rate of $1 \times 10^{-3} s^{-1}$ at room temperature using a Zwick/Roell tensile stage. The local strain ($\varepsilon_{Loc}$) evolution during the tensile test was evaluated by the digital image correlation (DIC) method with an Aramis system (GOM GmbH). The same tensile stage equipped with an additional furnace was used for the tensile test at high temperatures. The tensile specimens with the same size as the room temperature were heated through thermal convection. A thermocouple connected the furnace and the specimens was applied to control temperature. The stress and strain curves were measured by a force-sensing socket and displacement sensor. Argon atmosphere was used to minimize surface oxidation. Two minutes of holding were applied to achieve a homogeneous temperature distribution before mechanical loading. At least three samples were tested for each condition to confirm reproducibility. The Vickers hardness (Hv) was measured using a microhardness tester (LecoL-M100AT) with a Vickers diamond tip. A maximum load of $500g$ with a holding time of 13 s was performed on the polished surface for all the specimens. The distance between two indents is 300 μm to exclude the mutual impact. At least ten indents were tested for each material condition and the data points were averaged for each sample.

### Magnetic measurements

The magnetic properties of the investigated MCAs were measured in a quantum design physical properties measurement system (PPMS) equipped with a vibrating sample magnetometer (VSM) in the open-circuit condition using cuboid specimens with a dimension of $3 \times 3 \times 1 mm^3$, as shown in Fig. S3b. The hysteresis loop $M(H)$ measurements were extended to ±10,000 Oe at 5 K, 300 K, 573 K, 673 K, 773 K, and 873 K, respectively. The magnetization $M(T)$ temperature dependency was performed from 300 K to 1000 K under a 100 Oe

applied magnetic field. At least three samples were tested for each condition. The demagnetization effects stemming from the inner magnetic field in the sample are calculated by:

$$H_{int} = H_{appl} - N_d M \tag{1}$$

where $H_{int}$ is the internal or effective field, $H_{app}$ is the applied magnetic field strength, $N_d$ is the self-demagnetization correction factor, and $M$ is the volume magnetization of the sample under test. The $N_d$ of the specimens for PPMS measurement in a rectangular prism shape along the magnetic field direction ($z$-axis) is calculated using[37]:

$$
\begin{aligned}
\pi N_d = {} & \frac{b^2 - c^2}{2bc} \ln\left(\frac{\sqrt{a^2+b^2+c^2}-a}{\sqrt{a^2+b^2+c^2}+a}\right) + \frac{a^2-c^2}{2ac}\ln\left(\frac{\sqrt{a^2+b^2+c^2}-b}{\sqrt{a^2+b^2+c^2}+b}\right) \\
& + \frac{b}{2c}\ln\left(\frac{\sqrt{a^2+b^2}+a}{\sqrt{a^2+b^2}-a}\right) + \frac{a}{2c}\ln\left(\frac{\sqrt{a^2+b^2}+b}{\sqrt{a^2+b^2}-b}\right) + \frac{c}{2a}\ln\left(\frac{\sqrt{b^2+c^2}-b}{\sqrt{b^2+c^2}+b}\right) \\
& + \frac{c}{2b}\ln\left(\frac{\sqrt{a^2+c^2}-a}{\sqrt{a^2+c^2}+a}\right) + 2\tan^{-1}\left(\frac{ab}{c\sqrt{a^2+b^2+c^2}}\right) + \frac{a^3+b^3-2c^3}{3abc} \\
& + \frac{a^2+b^2-2c^2}{3abc}\sqrt{a^2+b^2+c^2} + \frac{c}{ab}\left(\sqrt{a^2+c^2}+\sqrt{b^2+c^2}\right) \\
& - \frac{(a^2+b^2)^{3/2}+(b^2+c^2)^{3/2}+(c^2+a^2)^{3/2}}{3abc}
\end{aligned}
\tag{2}
$$

where $a$, $b$, and $c$ are the width, thickness, and length of the specimens, respectively. The self-demagnetization correction factor is thus calculated to be 0.205. The potential effect of crystal anisotropy on the magnetic performance was investigated by measuring the hysteresis loop of the WP-MCA along a hot rolling longitudinal direction and transversal direction, as shown in Fig. S3a.

The static magnetic performance of the WP-MCA material was also obtained using a hysteresis curves (DC) test system (MAST-2010SD) under direct current (DC) conditions. The ring specimen with an outer diameter of 30 mm, an inner diameter of 25 mm, and a thickness of 3 mm was wound with copper wire for primary and secondary side windings. The specimen was designed to conform to the ASTM A773/A773M-01 standard. During the test, the specimen was immersed in water to prevent the perturbation caused by the thermal effect of the magnetizing current.

A digitally enhanced wide-field Kerr microscopy based on the magneto-optical Kerr effect (MOKE) was applied to measure magnetic hysteresis loops (MOKE magnetometry) and to image the evolution of magnetic domains during the magnetization reversal process[38]. Throughout the paper, the longitudinal Kerr effect was applied at orthogonal planes of incidence[39] to provide complementary sensitivity directions. The MOKE intensity of 16 image frames was averaged for each final image to reduce the noise. To enhance the domain image quality, the reference image in a saturated state was subtracted from the live view. To evaluate the temperature dependence of domain evolution, the sample was placed in a vacuum chamber with a glass window, allowing it to be heated to different temperatures without surface oxidation.

## Data availability

All data are available in the paper or the Supplementary Materials.

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

## Acknowledgements

L. Han would like to acknowledge the financial support from the China Scholarship Council (Number: 201906370028) and the data processing, analysis, and discussion with Dr. Jin Wang. The support of Dr. Dirk Ponge at the Max-Planck-Institut für Eisenforschung is gratefully acknowledged. IRSF acknowledges financial support through CAPES (Coordenação de Aperfeiçoamento de Pessoal de Nível Superior) & Alexander von Humboldt Foundation (grant number 88881.512949/2020-01). Z. Li would like to acknowledge the financial support from the National Natural Science Foundation of China (grant No. 51971248) and the Natural Science Foundation of Hunan Province in China (grant No. 2021JJ10056). We also acknowledge funding by the Deutsche Forschungsgemeinschaft (DFG, German Research Foundation), Project ID No. 405553726-TRR 270 (project ID A01 & Z01).

## Author contributions

L.H. designed the research project; L.H., F.M., N.P., and I.S. characterized the alloys; L.H., F.M., N.P., I.S., and I.R.S.F. analyzed the data; L.H., Z.L., R.S.O.G., and D.R. conceptualized the paper. All authors contributed to the writing of the paper.

## Funding

## Competing interests

The authors declare no competing interests.
