## [Peer Review File · Nature Communications]

Strong and ductile high temperature soft magnets through
Widmanstätten precipitatesREVIEWER COMMENTS

Reviewer #1 (Remarks to the Author):

The authors have reported the simultaneous achievement of exceptional soft ferromagnetism and favorable mechanical properties in the FeCoNiTa multicomponent alloy with Widmanstätten precipitates (WP-MCA), even at high temperatures. Structural, magnetic, and mechanical characterizations were conducted to analyze the alloy. Furthermore, a comparison was made with homogenized FeCoNiTa without Widmanstätten precipitates (HM-MCA). The authors propose that magnetic materials containing Widmanstätten precipitates have the potential to exhibit excellent soft ferromagnetism and high strength at elevated temperatures, attributed to the distinctive microstructure associated with these precipitates. The authors performed a simulation of the phase diagram in the FeCoNiTa alloy and identified a candidate material within a specific temperature range. The simulated results were well validated using actual ingots, confirming that the outstanding soft ferromagnetism and high strength do not deteriorate with increasing temperature. The authors also investigated the origin of this high performance at elevated temperatures by carefully analyzing changes in microstructures.

As mentioned in the manuscript's introduction, the magnetic performance of many soft ferromagnetic materials tends to degrade at high temperatures. Additionally, the combination of high strength and soft ferromagnetism is rarely reported but highly sought after for various industrial applications. The authors propose a new alloy design that achieves both exceptional soft ferromagnetism and favorable mechanical properties, even at high temperatures. This manuscript represents a significant milestone in the research of soft ferromagnetic materials. The organization of the manuscript is commendable, and the newly presented information is valuable for publication in Nature Communications. However, prior to acceptance, I recommend that the authors address the following issues to improve the paper:

- (1) In the Thermo-Calc simulation, the authors present the hcp phase, which corresponds to the D019 phase. However, many readers may generally associate hcp with the A3 phase, which is different from the D019 phase. If hcp refers to the D019 phase, the authors should modify the notation accordingly. If hcp represents the A3 phase, the authors will lose the theoretical foundation of the alloy design, and therefore, should reconsider the results of the Thermo-Calc simulation.
- (2) On page 11, line 10 from the bottom, the authors mention that the full width at half maximum (FWHM) of the WP-MCA decreases by 4.3% compared to its room temperature value. While this statement accurately applies to the (200) peak in Figure S7e, the reduction in FWHM for the (111) peak in Figure S7f appears to be less pronounced. The authors should provide a more detailed explanation of this disparity.
- (3) On page 4, line 7 from the bottom, the terms WP-MCA and HM-MCA are introduced. However, it is not immediately clear what distinguishes these two alloys until the method section in the supplemental materials is read. The authors should enhance the explanation of these alloys for better clarity.
- (4) On page 4, the term "Widmanstätten pattern)" should be corrected to "Widmanstätten pattern" (remove the closing parenthesis).
- (5) In Figure 1d, the fcc and hcp words are barely visible. Please enhance the visibility of these words.
- (6) Figure 3a includes the DCS curve, but no explanation is provided in the main text. Please provide commentary on the results of the DCS curve. Additionally, the explanation of T1 and T2 is missing. What do these temperatures represent?
- (7) On page 14, line 3, "Fig.5 dII and Fig.5dIII" should be corrected to "Fig.4 dII and Fig. 4dIII", respectively.
- (8) On page 14, line 14, "773K)" should be corrected to "773K".

(9) On page 16, line 7, "yellow" should be replaced with "orange".

Reviewer #2 (Remarks to the Author):

The authors describe a new magnetic alloy, based on the concept of multi-principal element/high-entropy alloy design. In particular, the authors describe a new alloy that involves a Widmanstätten-type precipitate mechanism that aids in retention of soft magnetic properties and good mechanical properties over a notable temperature range. Overall, the reviewer finds this manuscript to be of the quality, impact and detail expected of publications in Nature Communications. Indeed, the methodology outlined here for achieving so-called "magnetically soft but mechanically tough materials", as stated in the perspectives piece by Easo George in 2022, is novel and likely creates several opportunities for follow on research in the content of magnetic alloy design. Furthermore, the retention of sufficiently soft magnetic properties at elevated temperatures, with comparisons to even the current state-of-art FeCo alloy, is noteworthy, especially when considering the poor mechanical properties of FeCo (and other conventional soft magnetic alloys). The reviewer has only minor comments that the authors should address before final publication. Below are considerations for the authors.

On page 5, after a long list of results from the microstructural characterization, the authors state "this reveals two coexisting phases' high thermal stability, thus suppressing further coarsening and chemical partitioning, due to negligible driving forces." This statement is unclear to the reviewer – can the authors elaborate on this in the context of driving forces for microstructural evolution? The preceding discussion did not clarify the stability argument.

Would it be possible to plot properties of the current alloy, capturing the strength and maybe one measure of magnetic performance, as a function of temperature with other conventional magnetic alloys superimposed? While the text discusses some comparisons, for instance with conventional FeCo binary magnetic alloys, it may be more instructive to the reader to see these represented in an Ashby-style plot.

Regarding the magnetic properties characterization, the reviewer notes the use of a PPMS system with VSM. For this technique, while saturation/full-field induction values are routinely acquired as a function of temperature, structure-sensitive magnetic properties (e.g., coercivity, permeability, etc.) are sometimes difficult to ascertain with accuracy due to demagnetization effects driven by the free surfaces of the specimen geometry ends. This is especially true for thicker (i.e., non thin-film) sample geometries. Can the authors please provide more information on demag correction factors and approaches to account for these effects in this study, especially for coercivity? Further, since the alloy was cast into an ingot of 1-kg weight, the reviewer is wondering why a more typical B-H ring or yoke-type magnetic testing approach was not applied, at least at room temperature? This would avoid free-end and demag effects for a bulk material. Can the authors extract a B-H ring, in accordance with ASTM A773, to determine quasistatic magnetic properties to include in this study?

In the discussion, the authors mention crystal symmetry for the alloy phases. The reviewer is wondering what effects crystal anisotropy might have on the corresponding magnetic properties. In the case of conventional FCC lattices (the matrix of the present alloy), the easy magnetization direction is typically $\langle 111 \rangle$. Do the authors anticipate any crystallographic texture effects on their measured magnetic properties in this study? Was there any texturing that resulted from the sample preparation that should be noted? Similarly, the reviewer was curious what the rationale was for the sample preparation method, i.e., why was a 50% rolling reduction applied prior to heat treatment? Would there be any additional advantage of applying a smaller or greater rolling reduction in terms of the corresponding microstructure?

Finally, the reviewer notes the distinct ferromagnetic phases in the material. Aside from the D019 phase, was there evidence in any samples of chemical ordering, such as for the FCC matrix phase? Given that many conventional magnetic alloys, e.g., FeSi, FeCo, FeNi, present with long-range

ordering, the reviewer is curious if this was also observed in the present alloy. If not, is there a reason for retaining a chemically disordered FCC lattice?

Response to Reviewers' Report

Manuscript Title: Strong and ductile high temperature soft magnets through Widmanstätten precipitates

Manuscript number: NCOMMS-23-22606

We thank the reviewers for the valuable suggestions and comments on our manuscript. Our response is structured as follows: The comments from the reviewers are copied below in black and italic font. For each comment, we present a response item and the corresponding manuscript modifications (blue font). The changes in the amended manuscript are highlighted in yellow.

Reviewer Comments

Referee #1:

The authors have reported the simultaneous achievement of exceptional soft ferromagnetism and favorable mechanical properties in the FeCoNiTa multicomponent alloy with Widmanstätten precipitates (WP-MCA), even at high temperatures. Structural, magnetic, and mechanical characterizations were conducted to analyze the alloy. Furthermore, a comparison was made with homogenized FeCoNiTa without Widmanstätten precipitates (HM-MCA). The authors propose that magnetic materials containing Widmanstätten precipitates have the potential to exhibit excellent soft ferromagnetism and high strength at elevated temperatures, attributed to the distinctive microstructure associated with these precipitates. The authors performed a simulation of the phase diagram in the FeCoNiTa alloy and identified a candidate material within a specific temperature range. The simulated results were well validated using actual ingots, confirming that the outstanding soft ferromagnetism and high strength do not deteriorate with increasing temperature. The authors also investigated the origin of this high performance at elevated temperatures by carefully analyzing changes in microstructures.

As mentioned in the manuscript's introduction, the magnetic performance of many soft ferromagnetic materials tends to degrade at high temperatures. Additionally, the combination of high strength and soft ferromagnetism is rarely reported but highly sought after for various industrial applications. The authors propose a new alloy design that achieves both exceptional soft ferromagnetism and favorable mechanical properties, even at high temperatures. This manuscript represents a significant milestone in the research of soft ferromagnetic materials. The organization of the manuscript is commendable, and the newly presented information is valuable for publication in Nature Communications. However, prior to acceptance, I recommend that the authors address the following issues to improve the paper:

Response: We are most grateful to the reviewer for the careful reading, the pertinent comments, and the clear recommendation for the publication of our paper in Nature Communications. We have carefully addressed each point proposed by the reviewer to improve the paper, as explained in detail below.

(1) In the Thermo-Calc simulation, the authors present the hcp phase, which corresponds to the D019 phase. However, many readers may generally associate hcp with the A3 phase, which

is different from the D019 phase. If hcp refers to the D019 phase, the authors should modify the notation accordingly. If hcp represents the A3 phase, the authors will lose the theoretical foundation of the alloy design, and therefore, should reconsider the results of the Thermo-Calc simulation.

Response: We highly appreciate this pertinent comment which helps us to improve the manuscript. We double-checked the calculated equilibrium phase diagram of the $\text{Fe}_{35}\text{Co}_{30}\text{Ni}_{30}\text{Ta}_5$ (at.%) alloy by thermodynamic software (Thermo-Calc). The results show that the previously marked “hcp” phase is of “D0_a” (Ni_3Ta -type) structure, as shown in Fig. R1. The D0_a phase is also regarded as an ordered structure based on the hexagonal crystal lattice, i.e., the D0₁₉ crystal structure. We used the “hcp” structure in our original version to give the readership a rational alloy design guideline. We agree with the reviewer that many readers may generally associate hcp with the A3 phase, which may lead to confusion. In the revised manuscript, as suggested by the reviewer, we replaced “hcp” with “D0_a” to avoid misleading and calibrate the text correspondingly.

Fig. R1. Calculated equilibrium phase diagram of the $\text{Fe}_{35}\text{Co}_{30}\text{Ni}_{30}\text{Ta}_5$ (at.%) alloy using the thermodynamics software Thermo-Calc (see Methods).

Modifications: Please see the revised figure and figure caption on page 6: “The D0_a structure is an ordered D0₁₉ structure based on a hexagonal crystal”.

Please see the revised text on page 4: “The simulation results (Fig. 1a) show that a D0_a phase nucleates from the face-centered cubic (fcc#1) phase below 1360 K for this alloy composition. The D0_a crystal structure is an ordered D0₁₉ structure based on a hexagonal crystal lattice.”

“Fig. 1b shows the X-ray diffraction (XRD) patterns of the WP-MCA, with two families of diffraction peaks (face-centered cubic, hexagonal crystal).”

(2) On page 11, line 10 from the bottom, the authors mention that the full width at half maximum (FWHM) of the WP-MCA decreases by 4.3% compared to its room temperature value. While this statement accurately applies to the (200) peak in Figure S7e, the reduction in FWHM for the (111) peak in Figure S7f appears to be less pronounced. The authors should provide a more detailed explanation of this disparity.

Response: We appreciate the suggestion and fully agree. As pointed out by the reviewer, the original statement applies to the (200) peaks for the WP-MCA material at 773 K, while the reduction in FWHM for the (111) peaks is less pronounced. The disparity in the WP-MCA material could be due to the additional peaks near the (111) peaks from the hexagonal crystal structure, making the fitting challenging. In comparison, the fitting of the (200) peaks is unaffected. We followed the advice and provided a more detailed explanation of this disparity.

Modifications: Please see the revised manuscript, page 11: “More specifically, the full width at half maximum (FWHM) value of the (200) peak in the alloy WP-MCA at 773 K decreased by 4.3% compared to its room temperature value and is two times larger than that for the HM-MCA material.”

Please see the revised caption of Figure S9: “The decrease in full width at the half maximum value of the HM-MCA material from 773 K to 300 K are 2.1% and 1.6% for the (200) and (111) peaks by fitting, respectively, whereas the values for the WP-MCA material are 4.3 and 1.8%, respectively. The additional reflections near the (111) peak from the hexagonal crystal structure make the fitting challenging than that for the (200).”

(3) On page 4, line 7 from the bottom, the terms WP-MCA and HM-MCA are introduced. However, it is not immediately clear what distinguishes these two alloys until the method section in the supplemental materials is read. The authors should enhance the explanation of these alloys for better clarity.

Response: We thank the reviewer for the careful reading of our manuscript and the kind suggestions. We have fully complied and clarified the terms “WP-MCA” and “HM-MCA” in the revised manuscript.

Modifications: Please see the revised sentence on page 4: “We synthesized the homogenized multicomponent alloy (denoted as HM-MCA) with a single fcc crystal structure by conventional hot-rolling and homogenization of the cast alloy ingot (see Methods). The Widmanstätten-patterned multicomponent alloy (referred to as WP-MCA) with well-controlled precipitate size and number density was achieved after further isothermal heat treatment of the HM-MCA at 1173 K for 2 h.”

(4) On page 4, the term "Widmanstatten pattern)" should be corrected to "Widmanstatten pattern" (remove the closing parenthesis).

Modifications: Please see the revised sentence on page 4: “The Widmanstätten-patterned multicomponent alloy (referred to as WP-MCA) with well-controlled precipitate size and number density was achieved after further isothermal heat treatment of the HM-MCA at 1173 K for 2 h.

(5) In Figure 1d, the fcc and hcp words are barely visible. Please enhance the visibility of these words.

Response: We thank the reviewer for the excellent suggestion. We increased the font size and changed the colour of the font to be more visible and improve readability.

Modifications: Please see the updated Fig. 1d below and the revised manuscript on page 6.

Fig. 1. Microstructure analysis. (a) Calculated equilibrium phase diagram of the $\text{Fe}_{35}\text{Co}_{30}\text{Ni}_{30}\text{Ta}_5$ (at.%) alloy using the thermodynamics software Thermo-Calc (see Methods). The D0_a structure is an ordered D0_{19} structure based on a hexagonal crystal lattice. The black dashed line indicates the chosen isothermal annealing temperature (1173 K). (b) XRD patterns (measured and simulated). The inset shows the enlarged segment between 45° and 60° (black dashed frame). (c) ECCI analysis showing the microscale Widmanstätten-patterned precipitates. (d) TKD phase map showing the identified phases. The yellow frame refers to the

region shown in (e) and S1C (Suppl.). (e) DF-TEM image taken using D0₁₉ reflection (fig. S1d). (f) Atomic-resolution STEM micrograph showing the incoherent phase boundary. Left side, fcc matrix; Right side, D0₁₉ precipitates. (g) Corresponding FFT pattern. Red, fcc matrix, green, D0₁₉ precipitates. (h) An APT sample before sharpening showing the 3D plate-shaped precipitates. (i) Reconstructed APT map showing the elemental distributions. The interface is highlighted with 25 at.% Fe. The plane view (right-top corner) shows a 2.5 nm-thick cross-sectional slice from the tip. (j) 1D compositional profiles along the arrow marked in the plane view.

(6) *Figure 3a includes the DCS curve, but no explanation is provided in the main text. Please provide commentary on the results of the DCS curve. Additionally, the explanation of T₁ and T₂ is missing. What do these temperatures represent?*

Response: We thank the reviewer for the great suggestion. We followed this advice and made the corresponding modifications.

Modifications: Please see the updated text in the revised manuscript on page 9: "...according to the thermomagnetic and differential scanning calorimetry (DSC) heating curves (Fig. 3a)", and the revised figure caption of Fig. 3: "(a) Thermomagnetic and DSC curves. T_{c1} and T_{c2} on the thermomagnetic curve of the WP-MCA denote the Curie temperatures (T_c) of the fcc matrix and D0₁₉ precipitates, respectively. T₁ and T₂ on the DSC curve show the exothermic and second-order effects, respectively."

(7) *On page 14, line 3, "Fig.5 dII and Fig.5dIII" should be corrected to "Fig.4 dII and Fig. 4dIII", respectively.*

Response: We cordially thank the reviewer for carefully reading our manuscript and the suggestions. We apologize for lacking uniformity in style in our original manuscript version.

Modifications: We thoroughly check the revised manuscript to avoid any typos. Examples are given below.

Please see the revised figure captions on page 14: "particularly at the phase boundaries (Fig. 4dII). With a further increase of ε to 50% (Fig. 4dIII)"

(8) *On page 14, line 14, "773K)" should be corrected to "773K".*

Modifications: Please see the revised text on page 14: "Further, we investigated the deformation substructures after tensile tests at higher temperatures, i.e., 673 K and 773 K."

(9) *On page 16, line 7, "yellow" should be replaced with "orange".*

Modifications: Please see the revised text on page 16: “The shearing of the precipitates is highlighted by green dashed lines, and orange arrows highlight the microcracks;”

Referee #2:

The authors describe a new magnetic alloy, based on the concept of multi-principal element/high-entropy alloy design. In particular, the authors describe a new alloy that involves a Widmanstätten-type precipitate mechanism that aids in retention of soft magnetic properties and good mechanical properties over a notable temperature range. Overall, the reviewer finds this manuscript to be of the quality, impact and detail expected of publications in Nature Communications. Indeed, the methodology outlined here for achieving so-called “magnetically soft but mechanically tough materials”, as stated in the perspectives piece by Easo George in 2022, is novel and likely creates several opportunities for follow on research in the content of magnetic alloy design. Furthermore, the retention of sufficiently soft magnetic properties at elevated temperatures, with comparisons to even the current state-of-art FeCo alloy, is noteworthy, especially when considering the poor mechanical properties of FeCo (and other conventional soft magnetic alloys). The reviewer has only minor comments that the authors should address before final publication. Below are considerations for the authors.

Response: We cordially thank the reviewer for the strong support. We agree with the reviewer that advanced functional (magnetic) materials with good mechanical properties to operate under harsh environments for the electrification of transport, industry, energy supply, and sustainable infrastructures open substantial potential opportunities for following research. We thank the reviewer for the great suggestions, which helped us improve the manuscript for a broader readership.

On page 5, after a long list of results from the microstructural characterization, the authors state “this reveals two coexisting phases’ high thermal stability, thus suppressing further coarsening and chemical partitioning, due to negligible driving forces.” This statement is unclear to the reviewer – can the authors elaborate on this in the context of driving forces for microstructural evolution? The preceding discussion did not clarify the stability argument.

Response: We thank the reviewer for the excellent suggestion. We understand that the original statement presented directly after the microstructural characterizations may lead to confusion. The exothermic effect (T_1) at 1141 K from the DSC measurement (Fig. 3a) indicates that the original alloy (HM-MCA) was not in equilibrium condition. However, after annealing at a temperature (1173 K) higher than the T_1 for 2 h, the system is approaching equilibrium due to the formation of thermodynamically stable $D0_{19}$ precipitates. In addition, the driving force for precipitation coarsening due to the decrease of the total interface energy is small. This is confirmed by the microstructural evolution of alloy variants after different annealing times at 1173 K, i.e., prolonging the annealing time to 6 h has no obvious effect on the precipitation size (fig. S11). We deleted the original statement on page 5 and revised the sentences accordingly.

Modifications: Please see page 14 of the revised manuscript: “The high thermal stability of the two coexisting phases in the WP-MCA material has also been confirmed by exposing the

materials at 1173~1473 K for a duration of up to 6 h (fig. S13). No obvious precipitation coarsening and hardness change have been observed.”

Would it be possible to plot properties of the current alloy, capturing the strength and maybe one measure of magnetic performance, as a function of temperature with other conventional magnetic alloys superimposed? While the text discusses some comparisons, for instance with conventional FeCo binary magnetic alloys, it may more instructive to the reader to see these represented in an Ashby-style plot.

Response: We are most grateful for this great suggestion. We fully agree with the reviewer that a plot comparing mechanical and magnetic performance as a function of the temperature of the current new multicomponent alloy to the conventional magnetic alloys is helpful and instructive for better understanding. As pointed out by the reviewer, we also tried to compare the values in the original manuscript. However, most reported work does not simultaneously include high-temperature mechanical and magnetic properties. Yet, we fully comply and compare the yield strength and magnetization induction of the current multicomponent alloys with several conventional magnetic materials as a function of temperature individually, as shown in Fig. R2 below and in the revised supporting material:

Fig. R2. Plots comparing the temperature dependence of the (a) yield strength and (b) saturation induction of the WP-MCA with several commercially available high-temperature SMMs. The temperature dependence of yield strength of the FeCo-2V¹ and FeCo-V-0.5Nb² are compared. The temperature dependence of saturation induction (B_s) of the conventional magnetic Ni, 80Ni20Fe, 60Ni40Fe, 50Ni50Fe, Fe and 60Co40Fe materials³ are compared.

This comparison shows that the yield strength values of the new WP-MCA material are significantly higher than the conventional FeCo-2V and FeCo-V-0.5Nb alloys from room temperature up to 873 K. In addition, the WP-MCA material shows remarkable resistance

against high-temperature softening compared to the precipitation-free HM-MCA. The saturation induction of the WP-MCA is comparable to 50Ni50Fe and higher than those of binary Ni-Fe soft magnetic variants with Fe content less than 50 percent. Future efforts on developing soft magnetic MCAs for high-temperature applications could target variants for higher saturation induction and lower coercivity while preserving their outstanding mechanical performance.

Modifications:

Please see the revised manuscript on page 7: “The saturation induction (B_s) of WP-MCA is 1.1 T at 773 K while the B_s value of the 90.7 wt.% Co alloy is between 0.8 T (923 K) and 1.2 T (523 K) at 773 K²⁶. We also compared the B_s value of the current MCAs with a wide range of commercial SMMs in an overview plot (fig. S4a).”

Please see the revised manuscript on page 13: “A diagram comparing the temperature dependence of the yield strength of the current WP-MCA with several commercially available high-temperature SMMs is shown in fig. S4b. Notably, the σ_y value of the WP-MCA is 3 times larger than that of the conventional FeCo-2V alloy which has a σ_y value of 204 MPa at 300 K and 183 MPa at 873 K, respectively.”

Please see the conclusion: “Future efforts on designing strong and ductile soft magnetic MCAs for high-temperature applications could target higher saturation induction and lower coercivity while preserving their outstanding mechanical performance.”

Regarding the magnetic properties characterization, the reviewer notes the use of a PPMS system with VSM. For this technique, while saturation/full-field induction values are routinely acquired as a function of temperature, structure-sensitive magnetic properties (e.g., coercivity, permeability, etc.) are sometimes difficult to ascertain with accuracy due to demagnetization effects driven by the free surfaces of the specimen geometry ends. This is especially true for thicker (i.e., non thin-film) sample geometries. Can the authors please provide more information on demag correction factors and approaches to account for these effects in this study, especially for coercivity?

Response: We thank the reviewer for the nice comment. Indeed, the PPMS equipment with the VSM option measures the magnetization under open-circuit conditions and cannot directly consider the internal demagnetizing field. We also fully agree with the reviewer that based on the sample geometry and the orientation of the applied magnetic field, the demagnetization effect that stems from the inner field in the sample could be notable. This will mainly influence the shape of the hysteresis loop, but the obtained values of the saturation magnetization (M_s)

and coercivity (H_c) are not affected. This is based on the internal magnetic field strength modified by the demagnetization factor under open-circuit conditions as⁴:

$$H_{\text{int}} = H_{\text{appl}} - N_d M \quad (1)$$

where H_{int} is the internal or effective field, H_{appl} is the applied magnetic field strength, N_d is the self-demagnetization correction factor, and M is the volume magnetization of the sample under test. The schematic illustration of the hysteresis loops with and without demagnetization correction is shown in Fig. R3.

Fig. R3. The schematic initial hysteresis loops under open-circuit conditions without demagnetization correction (red), and the hysteresis loops after demagnetization correction (blue).

Another disadvantage of PPMS is that it casts uncertainty on H_c when the coercivity is extremely low. Yet, this is not the case in the current work since the H_c of the WP-MCA and HM-MCA at all testing temperatures are larger than 5 Oe. Therefore, the M_s and H_c values achieved in the current work by the PPMS equipment are trustworthy and comparable since we use the same specimen size for all the materials measured at different temperatures. These are why we did not apply correction factors and approaches in the original manuscript. However, we agree with the reviewer that the demagnetization correction is essential and helpful for readers to understand the intrinsic magnetic hysteresis loop better. Therefore, we calibrated the demagnetization effects driven by the free surfaces of the specimen geometry based on equation (1). The N_d of the specimens for PPMS measurement in rectangular prisms along the magnetic field direction (Fig. R4a) is calculated by⁵:

$$\begin{aligned} \pi N_d = & \frac{b^2-c^2}{2bc} \ln \left(\frac{\sqrt{a^2+b^2+c^2}-a}{\sqrt{a^2+b^2+c^2}+a} \right) + \frac{a^2-c^2}{2ac} \ln \left(\frac{\sqrt{a^2+b^2+c^2}-b}{\sqrt{a^2+b^2+c^2}+b} \right) + \\ & \frac{b}{2c} \ln \left(\frac{\sqrt{a^2+b^2}+a}{\sqrt{a^2+b^2}-a} \right) + \frac{a}{2c} \ln \left(\frac{\sqrt{a^2+b^2}+b}{\sqrt{a^2+b^2}-b} \right) + \frac{c}{2a} \ln \left(\frac{\sqrt{b^2+c^2}-b}{\sqrt{b^2+c^2}+b} \right) + \frac{c}{2b} \ln \left(\frac{\sqrt{a^2+c^2}-a}{\sqrt{a^2+c^2}+a} \right) + \\ & 2 \tan^{-1} \left(\frac{ab}{c\sqrt{a^2+b^2+c^2}} \right) + \frac{a^3+b^3-2c^3}{3abc} + \frac{a^2+b^2-2c^2}{3abc} \sqrt{a^2+b^2+c^2} + \frac{c}{ab} (\sqrt{a^2+c^2} + \\ & \sqrt{b^2+c^2}) - \frac{(a^2+b^2)^{3/2}+(b^2+c^2)^{3/2}+(c^2+a^2)^{3/2}}{3abc} \quad (2) \end{aligned}$$

where a, b and c are the width, thickness and length of the specimens, respectively. The self-demagnetization correction factor is calculated to be 0.205. Fig. R4b shows the initial and corrected hysteresis loops. The corresponding enlarged views near the zero external and saturation fields in Fig. R4c and d show the intrinsic coercivity and saturation magnetization values, respectively. The results show that the demagnetization effects do not notably affect the achieved saturation magnetization (M_s) and coercivity (H_c) values in the present study.

Fig. R4. **a**, Hysteresis loop of the WP-MCA acquired along the longitudinal and transversal directions up to ± 10000 Oe at room temperature. **b**, Measured and corrected hysteresis loops of the WP-MCA along the longitudinal direction acquired up to ± 10000 Oe at room temperature. The right-bottom inset shows the schematic illustration of the specimen dimension for calculating self-demagnetization correction factors along the applied magnetic field direction in the PPMS. The enlarged views showing the values of **c**, Coercivity and **d**, Saturation magnetization are not affected by the shape of the hysteresis loop in the current work. **e**, Closed-

circuit measurements showing the magnetic flux density in response to the applied field. The left-top inset shows the schematic illustration of the specimen winding and experiment set-up.

Modifications: Please see the revised manuscript on page 7: “Fig. 2a and b show the saturation magnetization (M_s) and coercivity (H_c) values averaged from three hysteresis loop measurements under open-circuit conditions.” and “We corrected the demagnetization effect based on the shape of the specimens. The demagnetization slightly changes the shape of the loop but does not notably affect the values of M_s and H_c (fig. S3b-d).”

Please see the revised supporting material on page 4: “The magnetic properties of the investigated MCAs were measured in a Quantum Design Physical Properties Measurement System (PPMS) equipped with a vibrating sample magnetometer (VSM) under open-circuit conditions using cuboid specimens with a dimension of $3 \times 3 \times 1 \text{ mm}^3$, as shown in fig. S3b.” and “The demagnetization effects stemming from the inner magnetic field in the sample are calculated by:

$$H_{\text{int}} = H_{\text{appl}} - N_d M \quad (1)$$

where H_{int} is the internal or effective field, H_{app} is the applied magnetic field strength, N_d is the self-demagnetization correction factor, and M is the volume magnetization of the sample under test. The N_d of the specimens for PPMS measurement in rectangular prisms shape along the magnetic field direction (z -axis) is calculated by:

$$\begin{aligned} \pi N_d = & \frac{b^2 - c^2}{2bc} \ln \left(\frac{\sqrt{a^2 + b^2 + c^2} - a}{\sqrt{a^2 + b^2 + c^2} + a} \right) + \frac{a^2 - c^2}{2ac} \ln \left(\frac{\sqrt{a^2 + b^2 + c^2} - b}{\sqrt{a^2 + b^2 + c^2} + b} \right) + \\ & \frac{b}{2c} \ln \left(\frac{\sqrt{a^2 + b^2} + a}{\sqrt{a^2 + b^2} - a} \right) + \frac{a}{2c} \ln \left(\frac{\sqrt{a^2 + b^2} + b}{\sqrt{a^2 + b^2} - b} \right) + \frac{c}{2a} \ln \left(\frac{\sqrt{b^2 + c^2} - b}{\sqrt{b^2 + c^2} + b} \right) + \frac{c}{2b} \ln \left(\frac{\sqrt{a^2 + c^2} - a}{\sqrt{a^2 + c^2} + a} \right) + \\ & 2 \tan^{-1} \left(\frac{ab}{c\sqrt{a^2 + b^2 + c^2}} \right) + \frac{a^3 + b^3 - 2c^3}{3abc} + \frac{a^2 + b^2 - 2c^2}{3abc} \sqrt{a^2 + b^2 + c^2} + \frac{c}{ab} (\sqrt{a^2 + c^2} + \\ & \sqrt{b^2 + c^2}) - \frac{(a^2 + b^2)^{3/2} + (b^2 + c^2)^{3/2} + (c^2 + a^2)^{3/2}}{3abc} \end{aligned} \quad (2)$$

where a , b and c are the width, thickness and length of the specimens, respectively. The self-demagnetization correction factor is thus calculated to be 0.205.”

Further, since the alloy was cast into an ingot of 1-kg weight, the reviewer is wondering why a more typical B-H ring or yoke-type magnetic testing approach was not applied, at least at room temperature? This would avoid free-end and demag effects for a bulk material. Can the authors extract a B-H ring, in accordance with ASTM A773, to determine quasistatic magnetic properties to include in this study?

Response: We appreciate the pertinent advice. As pointed out by the reviewer and mentioned in the above section, PPMS can measure the hysteresis loop as a function of temperature in a

wide temperature range. The measured saturation magnetization and coercivity values are precise without correcting the demagnetization effects. These are the main reasons we did not measure the magnetic performance using a typical $B-H$ ring. However, we fully agree with the reviewer that we can measure the magnetic performance of the current multicomponent soft magnets at least at room temperature using a $B-H$ ring and compare it with the results achieved by PPMS. This is because applying an analytically determined self-demagnetization correction factor does not always yield accurate corrections in practice. A magnetic circuit in the closed-circuit condition is ideally free of the self-demagnetization fields and associated corrections compared to the open-circuit condition. We fully comply.

Modifications: Please see the revised manuscript on page 7: “The magnetic performance of the WP-MCA at room temperature (300 K) was also measured under closed-circuit conditions (fig. S3e). The results are comparable with the values achieved in an open-circuit condition experiment, indicating that the values achieved from the VSM measurement are precise.”

Please see the revised supporting material on page 4: “The static magnetic performance of the WP-MCA material was also obtained using a hysteresis curves (DC) test system (MAST-2010SD) under direct current (DC) conditions. The ring specimen with an outer diameter of 30 mm, an inner diameter of 25 mm and a thickness of 3 mm was wound with copper wire for primary and secondary side windings. During the test, the specimen was immersed in water to prevent the perturbation caused by the thermal effect of the magnetizing current.

In the discussion, the authors mention crystal symmetry for the alloy phases. The reviewer is wondering what effects crystal anisotropy might have on the corresponding magnetic properties. In the case of conventional FCC lattices (the matrix of the present alloy), the easy magnetization direction is typically $\langle 111 \rangle$. Do the authors anticipate any crystallographic texture effects on their measured magnetic properties in this study? Was there any texturing that resulted from the sample preparation that should be noted?

Response: This is indeed an interesting point that deserves more discussion. The reviewer is correct that the easy magnetization direction in the fcc lattice is typically $[111]$, but for single crystals. The easy axis of magnetization of polycrystalline materials varies in individual crystals and is determined by the crystal orientations and other factors such as stress and temperature. In the current work, the multicomponent magnets are polycrystalline materials. We first processed the materials by conventional hot-rolling, followed by homogenization at 1473 K and annealing at 1173 K. The grains after homogenization show no preferred orientation and are free from residual stress, as shown in the electron backscatter diffraction (EBSD) analysis in Fig. R5. In addition, we prepared the specimen along the hot rolling longitudinal direction (RD) and transverse direction (TD) by electrical discharge machining to confirm this. We

measured the hysteresis loop by PPMS at room temperature, and the results are shown in Fig. R4a. The hysteresis loops in both directions show similar shapes and sizes. Therefore, crystallographic texture has no obvious effect on bulk performance.

Fig. R5. EBSD inverse pole figure map showing the equiaxed grains of the fcc matrix in the WP-MCA and the corresponding KAM map showing negligible misorientation value.

Modifications: Please see the revised manuscript on page 7: “The current WP-MCA material shows isotropic magnetic performance (fig. S3a) due to the polycrystalline grain structure.” and revised supporting material on page 4: “The potential effect of crystal anisotropy on the magnetic performance was investigated by measuring the hysteresis loop of the WP-MCA along hot rolling longitudinal direction (RD) and transverse direction (TD), as shown in fig. S3a.”

Similarly, the reviewer was curious what the rationale was for the sample preparation method, i.e., why was a 50% rolling reduction applied prior to heat treatment? Would there be any additional advantage of applying a smaller or greater rolling reduction in terms of the corresponding microstructure?

Response: We applied 50% engineering thickness reduction by hot rolling at ~1473 K above the recrystallization temperature. This is to eliminate the as-cast defects, e.g., shrinkage cavities and porosities, achieve uniformly distributed microstructure and prevent the materials from significant work hardening during processing. Also, the hot rolling reduction ratio can be tuned to achieve alloy sheets of the required thickness. For example, to meet the size of tensile specimens in flat shapes.

Finally, the reviewer notes the distinct ferromagnetic phases in the material. Aside from the D019 phase, was there evidence in any samples of chemical ordering, such as for the FCC matrix phase? Given that many conventional magnetic alloys, e.g., FeSi, FeCo, FeNi, present with long-range order, the reviewer is curious if this was also observed in the present alloy. If not, is there a reason for retaining a chemically disordered FCC lattice?

Response: We characterized the chemical ordering of the fcc matrix phase at the near-atomic scale by three-dimensional atom probe analysis, as shown in Fig. R6. The average Pearson coefficient (μ) for all the elements in the WP-MCA and the alloy variant after annealing at 773 K for 10 h is close to 0.1, suggesting a nearly random distribution of all elements. It should be noted that $\mu=0$ refers to complete randomness while $\mu=1$ indicates complete association in the occurrence of the solute atoms⁶. In addition, based on the high-resolution TEM analysis, the fcc phase does not show any detectable degree of long-range order at the near-atomic scale. One reason can be the sluggish kinetics of multicomponent alloys compared to conventional dilute alloys.

However, the values of μ in the alloy variant after further annealing the WP-MCA at 873 K for 10 h are notably increased, especially for Fe and Ta. More specifically, the μ of Fe increases more than 4 times from 0.099 ± 0.003 in the WP-MCA to 0.047 ± 0.012 after further annealing (10 h @ 873 K) MCA to 0.435 ± 0 . The μ of Ta increases 10 times from 0.047 ± 0.002 in the WP-MCA to 0.4827 after further annealing. The 1D compositional profiles further reveal the formation of the nanoscale Fe-enriched segregations in the alloy after further annealing at 873 K for 10 h. These results indicate the absence of local chemical fluctuations in the WP-MCA. This is likely to be the primary reason for the increase of the saturation magnetization, from $82.1 \text{ Am}^2/\text{kg}$ (WP-MCA at 873 K) to $95 \text{ Am}^2/\text{kg}$ (with prolonged annealing time to 10 h at 873 K). This is similar to conventional magnetic materials such as FeCo and Ni_3Fe . The effect of different degrees of ordering on the magnetic and mechanical properties has not been thoroughly investigated and explained, which will be our future interest.

Fig. R6. Pearson coefficient analysis of the WP-MCA and the alloy variants under further annealed conditions (e.g., 873 K for 10 h and 773 K for 10 h), respectively.

Modifications: We updated the APT analysis (see fig. S5) in the revised supporting file:“

Fig. S5.

APT analysis of the MCAs after further exposure at high temperatures. (a) Reconstructed APT map showing the elemental distributions after further annealing the WP-MCA at 873 K for 10 h. The corresponding plane view of a 2.5 nm-thick slice from the tip is shown. The interface is highlighted with 40 at.% Fe. It should be noted that the needle-like isosurface is the depleted artificial region, indicating typical crystallographic pole structures⁵⁴. **(b)** Reconstructed APT map of the MCA after annealing the WP-MCA at 773 K for 10 h and the

corresponding plane view of a 2.5 nm-thick slice from the tip. The concentration of Fe is lower than 40 at.% in the whole tip. (c) and (d) 1D compositional profiles from the rectangular region of interest along the black arrows in (a) and (b), respectively. (e) Average normalized homogenization parameter analysis.”

Please see the revised manuscript on page 9: “This can be attributed to the formation of the nanoscale Fe-enriched segregations observed to occur during annealing, as revealed by the elemental distribution and statistical deviation analysis (fig. S5).”

References

1. P. Moine, J. P. Eymery, P. Grosbras, *Physica status solidi (b)*. **46**, 177–185 (1971).
2. Ren, L., Basu, S., Yu, R.-H., Xiao, J. Q. & Parvizi-Majidi, A. Mechanical properties of Fe-Co soft magnets. *J Mater Sci* **36**, 1451–1457 (2001).
3. R. M. Bozorth, *Ferromagnetism*. (1993).
4. V. Franco, B. Dodrill, *Magnetic Measurement Techniques for Materials Characterization*. (Springer) (2021).
5. Aharoni, *J Appl Phys* **83**, 3432–3434 (1998).
6. M. P. Moody, L. T. Stephenson, A. V. Ceguerra, S. P. Ringer, *Microsc Res Tech* **71**, 542–550 (2008).

REVIEWERS' COMMENTS

Reviewer #1 (Remarks to the Author):

The authors have properly addressed all issues.
I recommend publication.

Reviewer #2 (Remarks to the Author):

The reviewer appreciates the efforts by the authors to address comments. The reviewer now recommends this manuscript for publication.